# Balance of Drug Residence and Diffusion in Lacrimal Fluid Determine Ocular Bioavailability in In Situ Gels Incorporating Tranilast Nanoparticles

**DOI:** 10.3390/pharmaceutics13091425

**Published:** 2021-09-08

**Authors:** Misa Minami, Hiroko Otake, Yosuke Nakazawa, Norio Okamoto, Naoki Yamamoto, Hiroshi Sasaki, Noriaki Nagai

**Affiliations:** 1Faculty of Pharmacy, Kindai University, 3-4-1 Kowakae, Higashi-Osaka, Osaka 577-8502, Japan; 2033420004w@kindai.ac.jp (M.M.); hotake@phar.kindai.ac.jp (H.O.); 2Faculty of Pharmacy, Keio University, 1-5-30 Shibakoen, Minato-ku, Tokyo 105-8512, Japan; nakazawa-ys@pha.keio.ac.jp; 3Okamoto Eye Clinic, 5-11-12-312, Izumi-Cho, Suita, Osaka 564-0041, Japan; eyedoctor9@msn.com; 4Center for Clinical Trial and Research Support, Research Promotion and Support Headquarters, Fujita Health University, 1-98 Dengakugakubo, Kutsukake, Toyoake 470-1192, Japan; naokiy@fujita-hu.ac.jp; 5Department of Ophthalmology, Kanazawa Medical University, 1-1 Daigaku, Uchinada, Kahoku 920-0293, Japan; mogu@kanazawa-med.ac.jp

**Keywords:** nanoparticles, tranilast, in situ gelling system, pluronic F-127, ophthalmic delivery

## Abstract

We previously designed ophthalmic formulations (nTRA) containing tranilast nanoparticles (Tra-NPs) with high uptake into ocular tissues. In this study, we used in situ gel (ISG) bases comprising combinations of pluronic F127 (F127) and methylcellulose (MC/F127), pluronic F68 (F68/F127), and Carbopol (Car/F127), and we developed in situ gels incorporating Tra-NPs (Tra-NP-incorporated ISNGs) such as nTRA-F127, nTRA-MC/F127, nTRA-F68/F127, and nTRA-Car/F127. Moreover, we demonstrated the therapeutic effect on conjunctival inflammation using lipopolysaccharide-induced rats. Each Tra-NP-incorporated ISNG was prepared by the bead mill method, the particle size was 40–190 nm, and the tranilast release and diffusion from formulation were nTRA > nTRA-F127 > nTRA-F68/F127 > nTRA-Car/F127 > nTRA-MC/F127. In the Tra-NP-incorporated ISNGs, the tranilast residence time in the lacrimal fluid, cornea, and conjunctiva was prolonged, although the *C*_max_ was attenuated in comparison with nTRA. On the other hand, no significant difference in conjunctival inflammation between non- and nTRA-F127-instilled rats was found; however, the nTRA-F68/F127, nTRA-Car/F127, and nTRA-MC/F127 (combination-ISG) attenuated the vessel leakage, nitric oxide, and tumor necrosis factor-α expression. In particular, nTRA-F68/F127 was significant in preventing the conjunctival inflammation. In conclusion, we found that the combination-ISG base prolonged the residence time of Tra-NPs; however, Tra-NP release from the formulation was attenuated, and the *T*_max_ was delayed longer than that in nTRA. The balance of drug residence and diffusion in lacrimal fluid may be important in providing high ocular bioavailability in formulations containing solid nanoparticles.

## 1. Introduction

Eyedrops provide many advantages, such as a noninvasive treatment adherence, compliance, self-administration, and reduced side effects [1,2]. Therefore, eyedrops are the most common formulations in the ophthalmic field. However, a large proportion of eyedrops is lost through nasolacrimal drainage, eye blinking, and binding to the surrounding extraorbital tissues [3,4,5], and the remaining eyedrops are also immediately diluted in the lacrimal fluid after instillation. As a result, pre-corneal and pre-conjunctival drug retention is poor, and less than 1–5% of the dose is delivered to the target tissues [1,6,7,8]. Therefore, frequent instillation is required to provide the desired therapeutic effects, although the repetitive instillation causes both topical and systemic side effects [9].

It is generally considered that enhancing pre-corneal and pre-conjunctival drug retention is the main approach to increasing the topical bioavailability (*BA*) and optimizing ocular drug delivery systems (DDSs). To improve traditional formulations, many researchers have attempted to develop ophthalmic DDSs to enhance the pre-corneal and pre-conjunctival drug retention. Solid nanoparticles (NPs), lipid NPs, polymers, liposomes, nanoemulsions, nanosuspensions, and micelles are some of the more successful additions to the formulation of eyedrops [10,11,12,13,14,15]. In particular, NPs have been used as ocular DDS, since the NPs enhance *BA* and reduce corneal toxicity [16,17]. We also developed solid tranilast (Tra) NPs (Tra-NPs) coated with 2-hydroxypropyl-β-cyclodextrin (HPβCD) and demonstrated that the resulting low corneal irritation and ocular *BA* in ophthalmic formulations containing Tra-NPs make them better than commercially available Tra eyedrops (CA-TRA) [18,19,20]. However, further improvements are expected to enhance the pre-corneal and pre-conjunctival retention to provide effective therapy for ophthalmic inflammation, such as allergic conjunctivitis.

To further enhance the contact with eye mucosa, various in situ gel (ISG) systems for ophthalmic treatment have been reported [21]. Eyedrops based on ISG systems convert to a gel upon ocular administration, resulting in improved pre-corneal and pre-conjunctival retention and drug *BA* [22,23]. Poloxamer (Pluronic^®^) is a block copolymer that consists of polypropylene oxide (PPO) and polyethylene oxide (PEO), and poloxamer at a certain concentration showed reverse thermal gelation with temperature [24,25]. Pluronic F127, which is a poloxamer, is liquid state in cold conditions (4 °C), but the viscosity is enhanced at body temperature, since the pluronic F127 undergoes a sol–gel transition at above 35 °C [26,27,28]. As a result of these characteristics, ophthalmic formulations with pluronic F127 can be administered in liquid form but turn into a gel after instillation, resulting in enhancement of the pre-corneal and pre-conjunctival retention and ocular *BA*. In addition, other ISG bases, such as methylcellulose (MC), pluronic F68, and Carbopol, are also used in the ophthalmic ISG system. MC and pluronic F68 exhibit thermoreversible gelation. On the other hand, Carbopol gels at the native value of the eye environment (pH 7.4). However, it needs high concentrations of an ISG base to enhance the pre-corneal and pre-conjunctival retention and drug *BA*, and the high content causes irritation in the ocular tissue. Recently, a combination of polymers was used in the DDS to decrease the total ISG content and improve the gelling characteristics, and several ocular formulations comprising combinations of pluronic F127 and MC (MC/F127), pluronic F68 (F68/F127), and Carbopol (Car/F127) were reported [29,30,31,32,33,34,35]. Based on these findings, designing an ophthalmic formulation with a combination of solid NPs and single or combination-ISGs may improve the pre-corneal and pre-conjunctival retention and ocular *BA* [29,30,31,32,33,34,35].

Tra has the ability to inhibit the TGF-β1 synthesis of various cells, resulting in inhibition of the accumulation of collagen in granulation tissue and the release of chemical mediators from the mast cell [36,37,38]. As a result of its efficacy, Tra has been widely and safely used as an anti-allergic drug in the ophthalmic field [39,40]. In this study, Tra-NPs were incorporated into various ISG bases (pluronic F-127 (10 and 15%), MC, pluronic F68, Carbopol) in combination to form an in situ gel (ISNG), and the Tra-NPs release from the in situ gel incorporating Tra-NPs (Tra-NP-incorporated ISNG) was characterized to evaluate the use of the ISG base for an ocular DDS. In addition, we investigated the preventive effect of Tra-NP-incorporated ISNGs on inflammatory mediators, such as nitric oxide (NO) and tumor necrosis factor-α (TNF-α), in lipopolysaccharide (LPS)-induced rat conjunctival inflammation [41,42].

## 2. Materials and Methods

### 2.1. Animals

Seven-week-old male Wistar rats, weighing 200–230 g, were commercially provided by the Kiwa Laboratory Animals Co., Ltd. (Wakayama, Japan). The rats were housed at ambient temperature with free access to a commercial diet (CR-3, Clea Japan Inc, Tokyo, Japan) and water. The 0.2 mg/mL LPS solution was prepared by dissolving in saline, and 30 µL of LPS solution was injected into the upper palpebral conjunctiva to induce the inflammation (LPS-induced rat). The inflammation model rats were used to evaluate the therapeutic effect of ophthalmic Tra formulations in this study. The protocol was conducted according to the guidelines of the Association for Research in Vision and Ophthalmology (ARVO) and Kindai University and approved by Kindai University (KAPS-31-003, 1 April 2019).

### 2.2. Chemicals

Pluronic F-127 was provided by Funakoshi Co., Ltd. (Tokyo, Japan), and Tra powder and CA-TRA were kindly donated by Kissei Pharmaceutical Co., Ltd. (Nagano, Japan). A Rat TNF-*α* Quantikine ELISA Kit was purchased from Bio-techne (Seattle, WA, USA), and fluorescein was obtained from Alcon (Tokyo, Japan). Pluronic F68, Evans blue (EB), isoflurane, ethyl p-hydroxybenzoate, and D-mannitol (Man) were purchased from Wako Pure Chemical Industries, Ltd. (Osaka, Japan). Benzalkonium chloride (BAC) was provided by Kanto Chemical Co., Inc. (Tokyo, Japan), and MC was purchased from Shin-Etsu Chemical Co., Ltd. (Tokyo, Japan). HPβCD and Carbopol (Carbopol^®^ 934) were obtained from Nihon Shokuhin Kako Co., Ltd. (Tokyo, Japan) and Serva (Heidelberg, Germany), respectively. Cell Count Reagent SF was purchased from Nacalai Tesque Inc. (Kyoto, Japan), and penicillin, Dulbecco’s Modified Eagle’s Medium/Ham’s F12, heat-inactivated fetal bovine serum (FBS), and streptomycin were provided by GIBCO (Tokyo, Japan). A Bio-Rad Protein Assay Kit was obtained from Bio-Rad Laboratories (Hercules, CA, USA). All other chemicals and organic solvents were of analytical grade.

### 2.3. Preparation of Tra-NP-Incorporated ISNG

Ophthalmic formulations containing Tra-NPs were prepared with a bead mill method based on our previous studies using a Shake Master NEO BMS-M10N21 (Bio-Medical Science Co. Ltd., Tokyo, Japan) [19,20,21,43,44]. Briefly, a mixture of BAC, Man, and Tra powder was added to the tube together with zirconia beads (diameter: 0.1 mm) and dispersed by the HPβCD solution. The dispersions were milled at 1500 rpm for 3 h with a Shake Master NEO BMS-M10N21, and Tra-NPs were prepared. The dispersions containing Tra-NPs were used as nTRA, and the Tra-NP-incorporated ISNGs (nTRA-F127, nTRA-MC/F127, nTRA-F68/F127, and nTRA-Car/F127) were prepared by the addition of pluronic F-127, MC/F127, F68/F127, or Car/F127 to nTRA, respectively. The compositions of the ophthalmic Tra formulations prepared in this study were determined according to previous reports [29,30,31,32] and are shown in Table 1. The pH of the ophthalmic formulation containing Tra-NPs was adjusted to 5.5, and sterilization was performed by filtration using a 0.22 µm pore membrane filter.

### 2.4. Characteristics of the Tra-NP-Incorporated ISNGs

The characteristics were evaluated following our previous study [18,19,21]. The mean particle sizes of Tra in mTRA and nTRA were measured by a laser diffraction particle size analyzer SALD-7100 (Shimadzu Corp. Kyoto, Japan), and the refractive index was set at 1.60–0.010i. Moreover, a Dynamic Light Scattering QuantumDesign NANOSIGHT LM10 (QuantumDesign Japan, Tokyo, Japan) was used to determine the particle size and number of the Tra-NPs in the ophthalmic formulations. The measurement was performed 5 times, and the mean was used as the particle size frequencies. An SPM-9700 (Shimadzu Corp., Kyoto, Japan) was used to provide a phase and height image, and the images were combined and shown as an atomic force microscopic (AFM) image. The zeta potential and viscosities were measured using a micro-electrophoresis zeta potential analyzer model 502 (Nihon Rufuto Co., Ltd., Tokyo, Japan) and an SV-1A (A&D Company, Limited, Tokyo, Japan), respectively. The Tra concentration was determined by the HPLC method using a Shimadzu HPLC LC-20AT system (Shimadzu Corp. Kyoto, Japan) with an Inertsil^®^ ODS-3 column (GL Science Co., Inc., Tokyo, Japan). Ethyl p-hydroxybenzoate was used as internal standard, and mobile phase (50 mM ammonium acetate and acetonitrile (80:20)) flowed at 0.25 mL/min. The Tra was detected at 230 nm at 35 °C, and the retention time is approximately 6.25 min (Appendix A) [18,19,20]. The soluble and non-solubilized Tra-NPs were separated by centrifugation at 100,000× *g* using a Beckman Optima^TM^ MAX-XP Ultracentrifuge (Beckman Coulter, Osaka, Japan), and Tra solubility was determined at 20 °C by the HPLC method described above. In this study, we attempted to investigate whether the aggregation and precipitation of Tra particles was observed for 1 month by using the dispersibility test. The dispersibility of ophthalmic formulations containing Tra-NPs was evaluated by the measurement of the Tra levels in the sample collected from the upper 90% of the tube over time. The total depth of the ophthalmic formulations containing Tra-NPs in the tube was 4 cm, and the formulations were incubated in the dark at 4 or 20 °C for 1 month.

### 2.5. Corneal Toxicity of Tra-NP-Incorporated ISNGs

The immortalized human corneal epithelial cell line HCE-T [45] was purchased from Araki-Sasaki and used for an in vitro cell tolerability assay, and cells were grown at 37 °C in a humidified atmosphere with 5% CO_2_. The growth medium had the following composition: DMEM-F12 with streptomycin (0.1 mg/mL) and penicillin (1000 IU/mL) fetal bovine serum, 5% *v*/*v*. A cytotoxicity test on the HCE-T cells was carried out using a Cell Count Reagent SF, which is a commercially available cell proliferation reagent. The HCE-T cells were plated at 1 × 10^4^ cells/well in 96-well microtiter plates; 24 h after plating, at 70% confluence, the growth medium was removed and replaced with 100 µL of the ophthalmic Tra formulations. In general, the eyedrops were diluted by the lacrimal fluid (approximately 10-fold dilution) immediately after the instillation, and drugs flowed through the nasolacrimal duct within 2–10 min. Taken together, the stimulation time was determined at 2 min according to the in vivo retention time of the drugs in the cornea [44]. After 2 min exposure (stimulation), the reaction medium was removed and washed twice with phosphate buffer, and 100 µL of fresh growth medium with 10 µL Cell Count Reagent SF was added in each well. After that, the HCE-T cells were incubated for 1 h at 37 °C to evaluate the cell damage, and the absorbance at 490 nm was measured using a microtiter reader. The background absorbance was measured on wells containing only the dye solution. In this study, the Abs of the non-treatment group was used as Control, and cell viability (%) was calculated as a ratio [Abs_treatment_/Abs_non-treatment_ (Control) × 100]. On the other hand, it is important to investigate the effect of prolonged drug residence time on the corneal toxicity. Therefore, we also evaluated the corneal toxicity in the in vivo study. The rats were used to evaluate the in vivo corneal toxicity of Tra-NP-incorporated ISNGs. The rats were repetitively instilled with ophthalmic Tra formulations three times a day (9:00, 15:00, and 21:00) for 2 months, the wound on the cornea was dyed with the instillation of 1% fluorescein and observed using a TRC-50X Fundus camera (Topcon, Tokyo, Japan), and the wound area was analyzed with image analyzing software Image J version 1.52.

### 2.6. Release of Tra from Tra-NP-Incorporated ISNGs

The in vitro release profile of Tra from the ophthalmic Tra formulations was studied using a methacrylate cell equipped with a 0.22 µm pore membrane filter [18,19,21,44]. The ophthalmic Tra formulations were added to the donor chamber of the methacrylate cell, and the other side (reservoir chamber) was filled with saline and incubated at 37 °C. Samples of 50 mL were taken from the reservoir chamber over time and replaced with the same volume of saline. The Tra levels in the sample were measured using the HPLC method described above. In addition, the particle number of Tra in the samples was evaluated using NANOSIGHT LM10, as described in detail elsewhere.

### 2.7. Tra Contents in Lacrimal Fluid, Blood, Cornea, and Conjunctiva

Seven-week-old male Wistar rats were single-instilled with ophthalmic Tra formulations and euthanized by the injection of a lethal dose of sodium pentobarbital 1, 3, and 5 h after instillation. Schirmer tear test strips were used to collect the lacrimal fluid, and both corneas and conjunctiva were excised. Thereafter, blood was sampled from the vena cava, and serum was provided by centrifugation at 20,400× *g* for 20 min at 4 °C. The Schirmer tear test strips, serum, corneas, and conjunctiva were homogenized in 100 µL methanol and centrifuged at 20,400× *g* for 20 min at 4 °C to extract Tra. The supernatants were used as samples, and the Tra contents in samples were analyzed by the HPLC method, as described in detail elsewhere. In this study, the Tra concentrations in the lacrimal fluid and blood are shown as mM and µM, respectively, and the Tra concentration in the corneas and conjunctiva are expressed as nmol/mg protein [18]. The protein levels in the samples were measured using a Bio-Rad Protein Assay Kit.

### 2.8. Vessel Leakage in Inflammation Using EB

The leakage of EB out of the vasculature in the conjunctiva was assayed to measure inflammation in LPS-induced rats instilled with or without ophthalmic Tra formulations. A total of 30 µL ophthalmic Tra formulation was instilled 30 min after LPS injection and left for 4 h. Then, the EB (10 mg/kg) was injected to the femoral vein, and the rats were euthanized by injecting a lethal dose of pentobarbital 1 h after EB injection. The blood was removed by perfusion with cold saline. Thereafter, the conjunctivas were excised and homogenized in 1 M KOH; then, they were incubated for 24 h at 37 °C. After that, 0.2 M phosphoric acid and acetone (5:13) was added into the homogenizer and incubated for 2 h to extract the EB; then, it was centrifuged at 400× *g* for 15 min at 4 °C. The EB levels in the supernatants were analyzed by the measurement of absorbance at 620 nm using the microplate reader MULTISKAN FC (Thermo Fisher Scientific, Kanagawa, Japan) and are expressed as Abs/g weight of conjunctival tissue [18].

### 2.9. Measurement of NO and TNF-α Levels

A total of 30 µL ophthalmic Tra formulation was instilled 30 min after 0.2 mg/mL LPS injection (30 µL) and left for 5 h. After that, the rats were euthanized by injecting a lethal dose of pentobarbital, and the conjunctivas were excised and homogenized in saline on ice; then, the supernatants were provided by centrifugation at 20,400× *g* for 20 min at 4 °C. The supernatants were used to measure the NO and TNF-*α* levels. The NO levels, which are the sum of the NO_2_^−^ and NO_3_^−^ levels, were measured by the Griess method using the ENO-20 (Eicom, Kyoto, Japan) connected to a concentric microdialysis probe (Eicom, Kyoto, Japan). Briefly, the supernatants were filtrated by a microdialysis probe (A-1-20-05) and perfused to the ENO-20 with Ringer’s solution (2 µL/min). Then, the samples were mixed with Griess reagent in the ENO-20, and the NO_2_^−^ and NO_3_^−^ levels were detected at 540 nm. Otherwise, the TNF-*α* levels were measured using a Rat TNF-*α* Quantikine ELISA Kit according to the manufacturer’s instructions. The TNF-*α* levels are expressed as pg/mg protein, and the protein levels in the samples were measured using a Bio-Rad Protein Assay Kit [18].

### 2.10. Statistical Analysis

All the experiments were carried out with at least five experiments per experimental point, and the data are presented as the mean ± standard error of the mean (S.E.M). Differences were assessed by ANOVA followed by Student’s *t*-test and Dunnett’s multiple comparisons, and *p* < 0.05 was considered as statistically significant.

## 3. Results

### 3.1. Evaluation of Physical Properties in the Tra-NP-Incorporated ISNGs

First, we attempted to design the ophthalmic formulations containing Tra-NPs (nTRA) by using the bead mill method, and Figure 1 shows the particle size and AFM images of nTRA. The particles of Tra without the bead mill treatment were in the micro-sized range (mean particle size 51.3 ± 0.15 µm, Figure 1A). On the other hand, the particle size of Tra was decreased by the bead mill treatment to the nano-sized range, and the particle size was 40–190 nm (Figure 1B–D). Moreover, the Tra-NPs had a shape close to spherical. Next, we prepared the Tra-NP-incorporated ISNGs using the pluronic F-127, which is a polymer with thermoresponsive behavior, and other ISG bases (MC, F68, and Car). The additives are important in the process of the bead mill, and the particle size of TRA was not nano-sized when the TRA was milled with a mixture of ISG base. Therefore, Tra-NP-incorporated ISNGs were prepared by the addition of pluronic F-127, MC/F127, F68/F127, or Car/F127 to nTRA, respectively. The particle size frequencies of Tra in the Tra-NP-incorporated ISNGs were not significantly different in comparison with nTRA. On the other hand, the viscosity in the ophthalmic Tra formulations was enhanced by the addition of an ISG base (pluronic F-127, MC/F127, F68/F127, or Car/F127), and the viscosities of nTRA-F127-L, nTRA-MC/F127-L, nTRA-F68/F127-L, and nTRA-Car/F127-L were 4.9-, 6.6-, 6.3-, and 6.4-fold higher, respectively, than that of nTRA at 20 °C (Table 2). Moreover, the viscosity was enhanced with the concentration of the ISG base, and the viscosities of nTRA-F127-H, nTRA-MC/F127-H, nTRA-F68/F127-H, and nTRA-Car/F127-H were approximately 16, 76, 70, and 70 mPa∙s, respectively. In addition, gelation of the Tra-NP-incorporated ISNGs was observed at 37 °C, and viscosity was remarkably enhanced (Table 2). At 37 °C, the viscosities of nTRA-F127-L, nTRA-MC/F127-L, nTRA-F68/F127-L, and nTRA-Car/F127-L were 5.7, 12, 9.3, and 28.8 mPa∙s, respectively, and the viscosities of nTRA-F127-H, nTRA-MC/F127-H, nTRA-F68/F127-H, and nTRA-Car/F127-H were 59-, 108-, 96-, and 112-fold higher, respectively, than that of nTRA (Table 2). In addition, we evaluated the particle size, NP number, and solubility of Tra in the Tra-NP-incorporated ISNGs (Table 2 and Figure 2). The size, number, and form of the Tra-NPs were similar to nTRAs and Tra-NP-incorporated ISNGs, and the addition of an ISG base (pluronic F-127, MC/F127, F68/F127, Car/F127) enhanced the solubility of Tra. In particular, the addition of Car/F127 increased the solubility of Tra in comparison with other ISG bases (pluronic F-127, MC/F127, F68/F127); however, 84.7% of Tra existed as Tra-NPs in the nTRA-Car/F127-H.

### 3.2. Evaluation of Dispersibility in the Tra-NP-Incorporated ISNGs

It was known that the zeta potentials described the stability of suspensions, and the additives were also related to the dispersibility. Moreover, it was important to investigate the dispersibility of the ophthalmic formulations for practical applications. Therefore, we measured the zeta potentials of the Tra-NP-incorporated ISNGs (Table 2) and demonstrated whether the addition of an ISG base (pluronic F-127, MC/F127, F68/F127, Car/F127) affected the aggregation of Tra in the Tra-NP-incorporated ISNGs 1 month after preparation (Figure 3). The zeta potential in the nTRA was −55 mV, and the aggregation and precipitation of Tra-NPs were observed at 4 and 20 °C 1 month after preparation. The zeta potential approached 0 with the addition of an ISG base containing pluronic F-127, MC/F127, or F68/F127, and the ISG base containing Car/F127 increased the negative charge. On the other hand, the changes in zeta potentials in the Tra-NP-incorporated ISNGs did not affect the dispersibility, since no aggregation and precipitation of Tra-NPs were observed in all the Tra-NP-incorporated ISNGs shown in Table 1 at 4 °C 1 month after preparation, and the Tra concentration in the upper layer of nTRA-Car/F127-H pH-sensitive gels was also approximately 0.5%. Thus, the dispersibility of nTRA was improved by the addition of an ISG base at 4 °C. In contrast to the results at 4 °C, the Tra concentration in the upper layer of nTRA-MC/F127-H and nTRA-F68/F127-H was poor, and Tra-NPs were aggregated at 20 °C.

### 3.3. Effect of the Tra-NP-Incorporated ISNGs on Corneal Toxicity in HCE-T Cell and Rat Corneas

Figure 4 shows the corneal toxicity of the Tra-NP-incorporated ISNGs. The viability of HCE-T cells treated with nTRA was 70.8%, and the corneal toxicity was lower than that of CA-TRA (56.5%). The viability of HCE-T cells was similar to that of cells treated with Tra-NP formulations containing low and high ISG bases (pluronic F-127, MC/F127, and F68/F127), and the cell toxicity of these formulations was lower than that of CA-TRA. On the other hand, the corneal toxicity of nTRA-Car/F127 was significantly higher than that of CA-TRA (Figure 4A). In the in vivo study, the corneal wound was not observed 2 months after the repetitive instillation of CA-TRA, nTRA, and Tra-NP-incorporated ISNGs with a low ISG base, and the corneal wound was not also observed in the treatment of nTRA-MC/F127-H and nTRA-F68/F127-H. However, slight corneal damage was observed after the repetitive instillation of nTRA-Car/F127-H, and the wound area was 0.71 ± 0.13 mm^2^ (*n* = 8).

### 3.4. Release of Tra from Tra-NP-Incorporated ISNGs

Figure 5 shows the effect of an ISG base on drug diffusion and release in the Tra-NP-incorporated ISNGs at 37 °C, and Table 3 lists Tra particle sizes in the reservoir chamber 5 h after treatment of each Tra-NP-incorporated ISNG. The Tra-NPs shifted to the reservoir side from the donor side in the nTRA and Tra-NP-incorporated ISNGs, although the addition of pluronic F-127 attenuated the release of Tra from nTRA-F127. Moreover, the combination of pluronic F-127 and another ISG base (MC, F68 or Car) also attenuated the release of Tra from formulation, and the release of Tra from Tra-NP formulations containing an ISG base (MC/F127, F68/F127 and Car/F127) was lower than that of nTRA. The release of Tra from Tra-NP-incorporated ISNGs decreased with the content of the ISG base. In addition, Tra release from nTRA-F68/F127-L was higher in comparison with nTRA-MC/F127 and nTRA-Car/F127, and the release of Tra-NPs was strongly prevented by the addition of the high MC/F127 base.

### 3.5. Drug Behavior in Rat Eyes Instilled with Tra-NP-Incorporated ISNGs

Figure 6 and Figure 7 show the Tra contents in the lacrimal fluid (Figure 6A,B), blood (Figure 6C,D), cornea (Figure 7A,B), and conjunctiva (Figure 7C,D) of rats after the instillation of Tra-NP-incorporated ISNGs. The *C*_max_ of Tra in the blood of rats instilled with nTRA decreased with the addition of an ISG base, although the plasma Tra levels in the rats instilled with Tra-NP-incorporated ISNGs were higher than that of nTRA 5 h after the instillation. On the other hand, the addition of an ISG base increased and prolonged the retention time in the lacrimal fluid, cornea, and conjunctiva. Although the retention time of Tra in the lacrimal fluid was not significantly different between nTRA-MC/F127, nTRA-F68/F127, and nTRA-Car/F127, the Tra levels in the cornea and conjunctiva of rats instilled with nTRA-F68/F127 were significantly higher than those with nTRA-MC/F127 and nTRA-Car/F127. Moreover, the Tra levels in the cornea and conjunctiva of rats instilled with nTRA-MC/F127 were lower in comparison with nTRA-F68/F127 and nTRA-Car/F127, and the Tra levels decreased with the contents of the MC/F127 base.

### 3.6. Preventive Effect of the Instillation of Tra-NP-Incorporated ISNGs on Inflammation in the Conjunctiva

Figure 8 shows the EB exudation data (A), NO levels (B), and TNF-α levels (C) in the conjunctiva of conjunctivitis rats instilled with Tra-NP-incorporated ISNGs. The injection of LPS induced EB exudation and enhanced the NO and TNF-α levels in the conjunctiva. The instillation of CA-TRA, nTRA, and nTRA-F127 did not attenuate the EB exudation, NO levels, or TNF-α levels in the conjunctiva; however, the EB exudation in conjunctivitis rats was significantly prevented by the instillation of combination-ISGs incorporating Tra-NPs (nTRA-MC/F127, nTRA-F68/F127, and nTRA-Car/F127). The preventive effect of EB exudation with Tra-NP-incorporated ISNGs with a low ISG base was higher than that with Tra-NP-incorporated ISNGs with a high ISG base. Moreover, the NO and TNF-α levels in the conjunctiva were attenuated by the instillation of combination-ISGs incorporating Tra-NPs. The NO and TNF-α levels were not significantly different between Tra formulations containing low and high ISG bases, although the preventive effect of NO and TNF-α levels in the Tra-NP-incorporated ISNGs with a low ISG base tended to increase in comparison with Tra-NP-incorporated ISNGs with a high ISG base (Figure 8B,C).

## 4. Discussion

The aim of this study was to prepare not only solid NPs of Tra but also Tra-NP-incorporated ISNGs with various ISG bases (pluronic F127, MC, pluronic F68, Carbopol) in combination. Through research, we found that the increase in viscosity (gelation) prolonged the pre-corneal residence time of Tra-NPs; however, it attenuated the delay of *T*_max_ (speed of absorption) via the NPs release from formulation and showed that this balance was related to the *BA* in the cornea and conjunctiva. In addition, we mentioned that to increase the ocular *BA* of solid NPs with the ISNGs, it is important to optimize the concentration of the ISG base in the ophthalmic formulations (Figure 9).

First, we attempted to prepare the dispersions containing Tra-NPs with the bead mill method. In general, the BAC was used as a preservative in the ophthalmic formulation, and mannitol was added to prevent the ophthalmic stimulation of BAC [44]. In addition, our previous study showed that the HPβCD attenuated the aggregation of solid NPs in the dispersions [44]. From these findings, we selected three additives (BAC, mannitol, and HPβCD) to prepare the ophthalmic formulation containing Tra-NPs (nTRA). The Tra particles in nTRA were 40–190 nm (Figure 1), and we succeeded in preparing the dispersions containing solid Tra-NPs. On the other hand, no aggregation and precipitation of Tra-NPs was observed 6 d after preparation, although the aggregation and precipitation of Tra-NPs in the nTRA were observed at 1 month (Figure 2).

Next, we designed the Tra-NP-incorporated ISNGs by combining the bead mill method and the ISG base. The changes in temperature and pH affected the gelation of the in situ gelling polymers in the pre-corneal and pre-conjunctiva environment. Pluronic F127 is a thermoreversible gel [27], and the mechanism of gelation led to an increase in hydrophobicity by the rupture of hydrogen bonds of the PPO under the heating [46]. As a result of these characteristics, pluronic F127 is attractive in formulating thermoreversible gels in the ophthalmic formulation for the improvement of pre-corneal residence time, and pluronic F127 based on the ISG system containing various drugs has been used for treatment in the ophthalmic field [47,48]. In addition, it was reported that gelling temperature was affected not only by the pluronic F127 concentration but also by the proportions of combined pluronic F127 and pluronic F68 [49]. Moreover, temperature-sensitive MC and pH-sensitive Carbopol are also frequently used as agents when combining two ISGs [50,51]. We prepared the Tra-NP-incorporated ISNGs with the combination of pluronic F127 and MC, pluronic 68, Carbopol, and determined the concentration of the ISG base following previous reports for ISG systems [32,47,48,49,50]. The additives for ISG bases (Pluronics F127, MC/F127, F68/F127, and Car/F127) were not significantly affected by the particle size, and the particle size frequencies in each ophthalmic Tra formulation were approximately 70–210 nm (Figure 2). An aqueous solution of pluronic F127 in the concentration of 15% or higher is transformed into a semisolid gel above 25 °C. In this study, the gelation of Tra formulations containing 15% pluronic F127 was observed (Table 2). The viscosity of Tra formulations containing 10% pluronic F127 also increased, although the state was a viscous solution (Table 2). Although the dispersibility of nTRA improved with the addition of an ISG base at 4 °C, the aggregation and precipitation of Tra-NPs were observed at 20 °C in nTRA-MC/F127-H and nTRA-F68/F127-H (Figure 3). In addition, the changes in zeta potentials in the Tra-NP-incorporated ISNGs did not affect the aggregation and precipitation of Tra-NPs, since no aggregation and precipitation of Tra-NPs were observed in the Tra-NP-incorporated ISNGs at 4 °C 1 month after preparation. On the other hand, MC, pluronic F127, and pluronic F68 undergo reverse thermal gelation with temperature, and Carbopol is a pH-sensitive ISG base and gels under the pH 7.4. In this study, no gelation was observed in nTRA-MC/Car-H at 20 °C, since the pH of the formulation was the same at 4 and 20 °C; however, incomplete gelation was observed in nTRA-MC/F127-H and nTRA-F68/F127-H with thermal-sensitive gel at 20 °C. From these results and information, the aggregation and precipitation of Tra-NPs at 20 °C may be caused by the thermal gelation. The Tra formulations containing Car/F127 can be stored at room temperature, although Tra formulations containing high MC/F127 and F68/F127 should be stored under cool conditions (4 °C).

Furthermore, we demonstrated the corneal toxicity of Tra-NP-incorporated ISNGs using both HCE-T cells and rats. Corneal damage was observed after the treatment with nTRA-Car/F127-L (Figure 4). It has been reported that pH-sensitive gels are more toxic in comparison to temperature-sensitive gels, and pH-sensitive gels easily damage conjunctival cells [51]. Therefore, this result suggested that the corneal toxicity of nTRA-Car/F127 was caused by Carbopol and pH-sensitive gels, and caution is necessary with long-term use of nTRA-Car/F127-H. In contrast to nTRA-Car/F127, the repetitive instillation of Tra formulations containing MC/F127 and F68/F127 was safe.

In general, the high viscosity reduced the diffusion of Tra-NPs in the ophthalmic formulation. Therefore, we demonstrated the relationship between viscosity and diffusion in the Tra-NP-incorporated ISNGs (Figure 5). Although the Tra-NPs were released from nTRA-F127, the addition of an ISG base decreased the diffusion and release of Tra in the formulation with an ISG base. In the combination-ISGs incorporating Tra-NPs, the Tra-NP diffusion in nTRA-F68/F127 was higher than that in nTRA-MC/F127 and nTRA-Car/F127, and the Tra-NP diffusion in nTRA-Car/F127 was the lowest (Figure 5). In addition, we also measured the in vivo accumulation of Tra in ocular tissues after instillation. Although total Tra levels in the blood were not observed to be significantly different with the addition of an ISG base, the decrease in *C*_max_ and sustained blood concentration were observed (Figure 6C,D). Moreover, the combination-ISGs incorporating Tra-NPs prolonged the residence time of Tra content in the lacrimal fluid, cornea, and conjunctiva in comparison with rats instilled with nTRA-F127 (Figure 6 and Figure 7). On the other hand, the addition of an ISG base to the Tra-NP formulations provided the delay in *T*_max_ and decrease in *C*_max_ in the cornea and conjunctiva of rats (Figure 7). In addition, the *C*_max_ in the cornea and conjunctiva were increased with the Tra formulation, showing high Tra-NP diffusion. These results suggested that the increase in viscosity prolonged Tra-NP residence time on the ocular surface; however, it attenuated the speed of absorption. It is possible that this balance was related to absorption by the cornea and conjunctiva in the instillation of the Tra-NP-incorporated ISNG.

Clinically, Tra has been used in the treatment of allergic conjunctivitis, allergic rhinitis, and asthma. Therefore, it is important to evaluate the therapeutic effect of Tra-NP-incorporated ISNGs on conjunctival inflammation. In the previous study, LPS was used to induce an inflammatory response, since the LPS is a potent simulator of cytokine secretion via monocyte and macrophage [52,53,54]. Moreover, NO and TNF-α were also caused by the LPS-induced inflammation, and it is known that these inflammatory mediators play roles mainly in inflammatory processes [53,55]. In this study, we used the in vivo conjunctival inflammation model of the injection of LPS and investigated whether the Tra-NP-incorporated ISNGs suppressed the inflammation. No significant difference was found in conjunctival inflammation between control and nTRA-F127-instilled rats (Figure 8A). In contrast to the results with nTRA-F127, the combination-ISG incorporating Tra-NPs attenuated the vessel leakage, and the NO and TNF-α levels in the conjunctiva were also reduced by the instillation of combination-ISGs incorporating Tra-NPs (Figure 8B,C). Moreover, the preventive effects on vessel leakage by Tra-NP-incorporated ISNGs with a low ISG base were higher than those of Tra-NP-incorporated ISNGs with a high ISG base, and the NO and TNF-α levels in the conjunctiva also tended to decrease (Figure 8). We hypothesize that the combination-ISG base prolongs the pre-corneal and pre-conjunctival contact time of Tra-NPs. On the other hand, the high anti-inflammatory activity of Tra via enough drug uptake in the ophthalmic tissue provided some degree of viscosity, since the *C*_max_ in the cornea and conjunctiva of rats instilled with Tra-NP-incorporated ISNGs with a high ISG base was lower than that with Tra-NP-incorporated ISNGs with a low ISG base (Figure 8). Further studies are needed to clarify the usefulness of ISNGs based on a combination of ISGs and Tra-NPs with rabbits and monkeys. In addition, it is necessary to elucidate qualitative analysis at the cellular level of NPs in preventing the conjunctiva inflammation effectively and revealing the specific mechanism of anti-inflammation in the cornea.

## 5. Conclusions

We prepared Tra-NPs by the bead mill method and designed ISNGs based on a combination of ISGs and Tra-NPs. In addition, we found that the addition of an ISG base prolonged the pre-corneal and pre-conjunctival contact times of Tra-NPs with ISGs; however, the Tra-NPs’ release from formulations containing a high ISG base was attenuated, and the *T*_max_ (speed of absorption) was also delayed in comparison with formulations containing a low ISG base. Therefore, the optimal content of the ISG base is important to providing high *BA* in the cornea and conjunctiva. Our studies are the first to evaluate the relationship of viscosity and drug behavior after the instillation of Tra-NP-incorporated ISNGs.

## Figures and Tables

**Figure 1 pharmaceutics-13-01425-f001:**
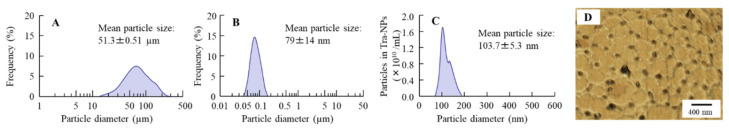
Particle size and atomic force microscopy (AFM) images of tranilast (Tra) in ophthalmic formulation. (**A**) and (**B**) Particle size frequencies of Tra in ophthalmic dispersions containing Tra-microparticles (**A**, mTRA) and Tra-nanoparticles (**B**, nTRA). The data were measured by SALD-7100. (**C**) Particle size frequencies of Tra-nanoparticles (Tra-NPs) in nTRA measured by NANOSIGHT LM10. (**D**) AFM images of Tra-NPs in nTRA. Compositions of mTRA and nTRA are shown in Table 1. The Tra particles in mTRA and nTRA were 15–300 µm and 40–190 nm, respectively. The difference in particle size in Figure 1B,C is due to the measuring method (Figure 1B, SALD-7100. Figure 1C, NANOSIGHT LM10).

**Figure 2 pharmaceutics-13-01425-f002:**
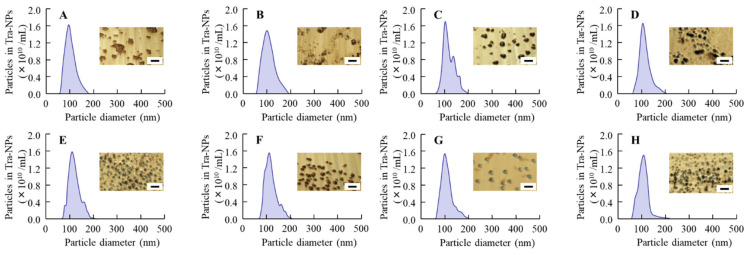
Particle size frequencies and AFM images of Tra-NPs in nTRA-F127-L (**A**), nTRA-F127-H (**B**), nTRA-MC/F127-L (**C**), nTRA-MC/F127-H (**D**), nTRA-F68/F127-L (**E**), nTRA-F68/F127-H (**F**), nTRA-Car/F127-L (**G**), and nTRA-Car/F127-H (**H**). Compositions of each Tra-NP-incorporated ISNG shown in Table 1. The bar in the AFM image indicates 200 nm. The ISG bases were not affected by the particle size, and the particle size frequencies in each Tra-NP-incorporated ISNG were approximately 70–210 nm.

**Figure 3 pharmaceutics-13-01425-f003:**
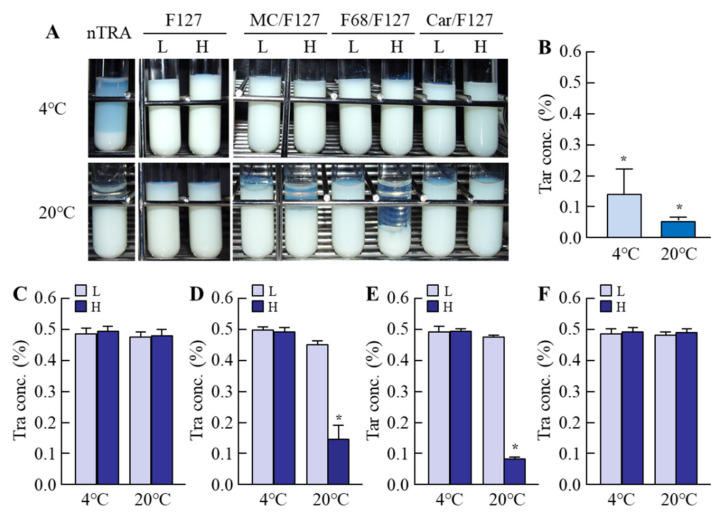
Changes in dispersion stability in each ophthalmic Tra formulation 1 month after preparation. (**A**) Images of dispersibility in each ophthalmic Tra formulation at 4 and 20 °C. (**B**–**F**) Effect of ISG bases on dispersion stability in the nTRA (**B**), nTRA-F127 (**C**), nTRA-MC/F127 (**D**), nTRA-F68/F127 (**E**), and nTRA-Car/F127 (**F**). The compositions of the ophthalmic Tra formulations are shown in Table 1. *n* = 5. * *p* < 0.05 vs. Tra con. immediately after preparation (0.5%). The low dispersibility of nTRA was improved by the addition of an ISG base at 4 °C, although the Tra concentration in the upper layer of nTRA-MC/F127-H and nTRA-F68/F127-H was significantly poor at 20 °C, and the aggregation and precipitation of Tra-NPs were observed.

**Figure 4 pharmaceutics-13-01425-f004:**
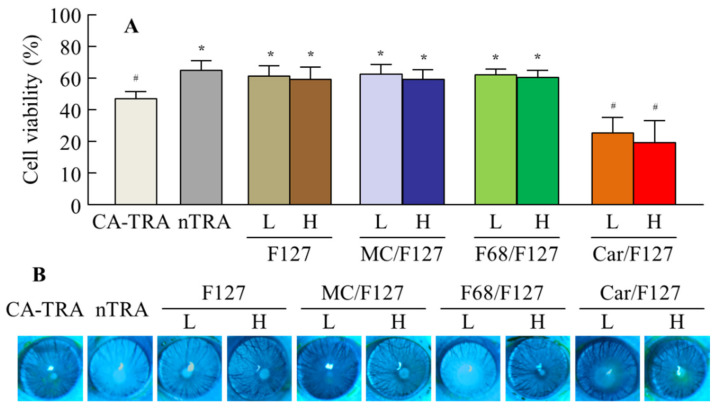
Corneal toxicity of each ophthalmic Tra formulation. (**A**) Changes in the cell viability of HCE-T cells treated with each ophthalmic Tra formulation. Each ophthalmic Tra formulation was applied for 2 min. (**B**) Effect of repetitive instillation on the rat cornea. The repetitive instillation was performed three times a day for 2 months. The compositions of the ophthalmic Tra formulations are shown in Table 1. *n* = 8. * *p* < 0.05 vs. CA-TRA for each category. ^#^ *p* < 0.05 vs. nTRA for each category. The corneal toxicities of nTRA, nTRA-F127, nTRA-MC/F127, and nTRA-F68/F127 were lower than that of CA-TRA. On the other hand, the cell was stimulated by the treatment with nTRA-Car/F127, and slight corneal damage was observed in rats repetitively instilled with nTRA-Car/F127-H.

**Figure 5 pharmaceutics-13-01425-f005:**
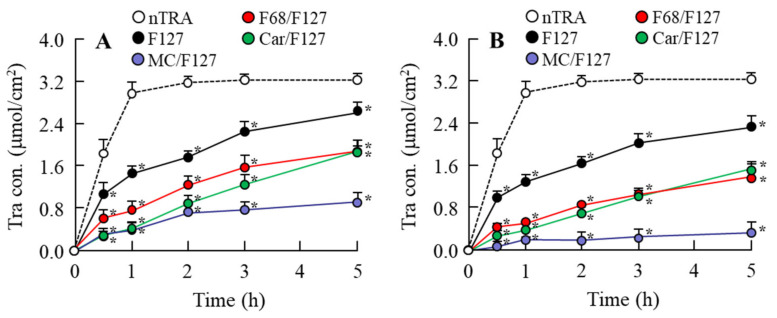
Diffusion of Tra in each ophthalmic Tra formulation through a methacrylate cell. (**A**) Diffusion behavior of Tra in the ophthalmic Tra-NP-incorporated ISNGs with a low ISG base. (**B**) Diffusion behavior of Tra in the ophthalmic Tra-NP-incorporated ISNGs with a high ISG base. The compositions of the ophthalmic Tra formulations are shown in Table 1. *n* = 6–9. **p* < 0.05 vs. nTRA for each category. The release of Tra from Tra-NP-incorporated ISNGs decreased with the content of the ISG base, and the Tra release with the combination of pluronic F-127 and another ISG base (MC, F68 and Car) was lower than that of nTRA. The release levels of Tra were nTRA > nTRA-F127 > nTRA-F68/F127 > nTRA-Car/F127 > nTRA-MC/F127.

**Figure 6 pharmaceutics-13-01425-f006:**
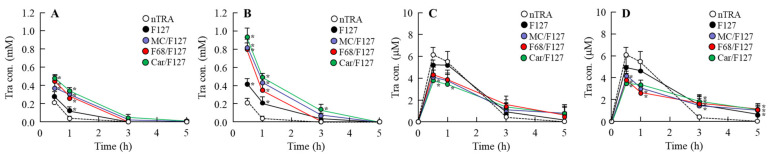
Changes in Tra contents in the lacrimal fluid and blood of rats instilled with each ophthalmic Tra formulation. (**A**,**B**) Tra behavior in the lacrimal fluid of rats instilled with the Tra formulations containing a low (**A**) and a high (**B**) ISG base. (**C**,**D**) Tra behavior in the blood of rats instilled with the Tra formulations containing a low (**C**) and a high (**D**) ISG base. The compositions of the ophthalmic Tra formulations are shown in Table 1. *n* = 5–10. * *p* < 0.05 vs. nTRA for each category. The residence time and content of Tra in the lacrimal fluid were enhanced by the addition of an ISG base. In the blood, prolonged residence time and decreased *C*_max_ were observed in the combination-ISGs incorporating Tra-NPs.

**Figure 7 pharmaceutics-13-01425-f007:**
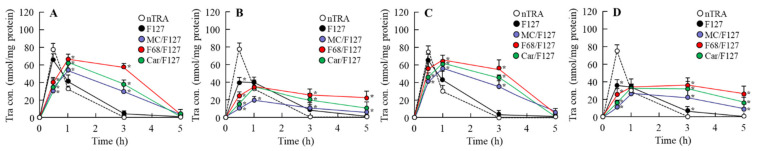
Changes in Tra contents in the cornea and conjunctiva of rats instilled with each ophthalmic Tra formulation. (**A**,**B**) Tra behavior in the cornea of rats instilled with the Tra formulations containing a low (**A**) and a high (**B**) ISG base. (**C**,**D**) Tra behavior in the conjunctiva of rats instilled with the Tra formulations containing a low (**C**) and a high (**D**) ISG base. The compositions of the ophthalmic Tra formulations are shown in Table 1. *n* = 5–10. * *p* < 0.05 vs. nTRA for each category. The residence time of Tra content in the cornea and conjunctiva was prolonged by the addition of an ISG base; however, the *C*_max_ in the Tra-NP-incorporated ISNGs were attenuated in comparison with nTRA.

**Figure 8 pharmaceutics-13-01425-f008:**
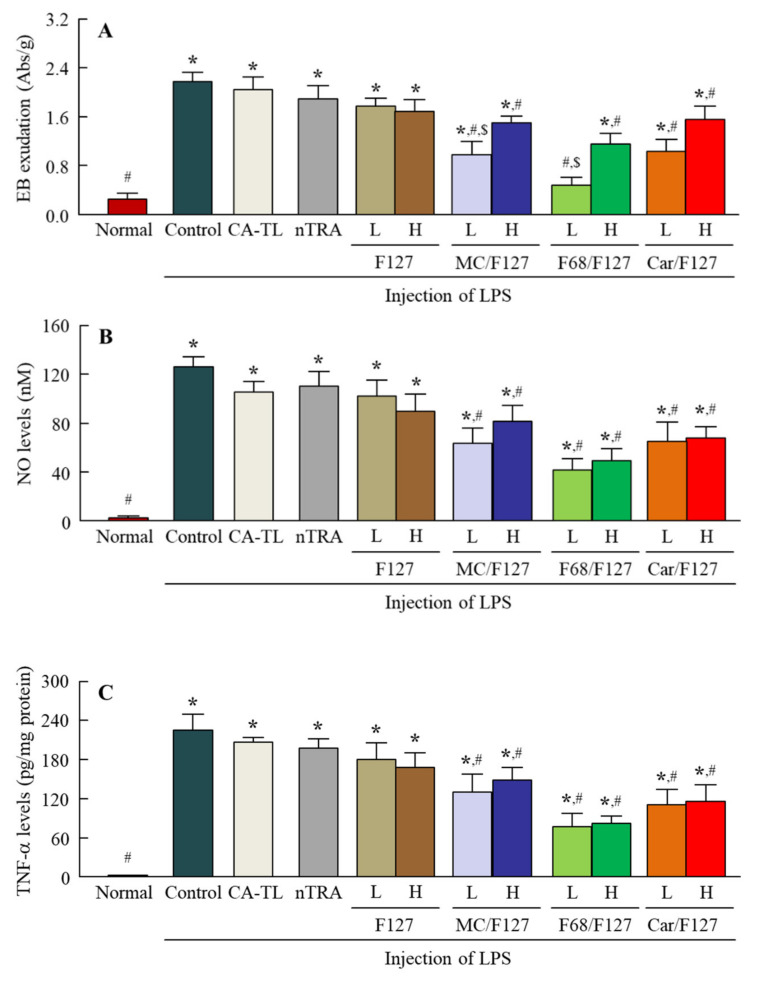
Changes in EB exudation (**A**), NO levels (**B**), and TNF-α levels (**C**) in the conjunctiva of conjunctivitis rats 5 h after the instillation of each ophthalmic Tra formulation. The compositions of the ophthalmic Tra formulations are shown in Table 1. *n* = 5–12. * *p* < 0.05 vs. Normal for each category. ^#^ *p* < 0.05 vs. Control for each category. ^$^ *p* < 0.05 vs. Tra-NP-incorporated ISNGs with a high ISG base for each category. The instillation of combination-ISGs incorporating Tra-NPs attenuated EB exudation, NO levels, and TNF-α levels caused by LPS injection. In particular, nTRA-F68/F127-L significantly prevented the EB exudation, NO levels, and TNF-α levels 5 h after the instillation.

**Figure 9 pharmaceutics-13-01425-f009:**
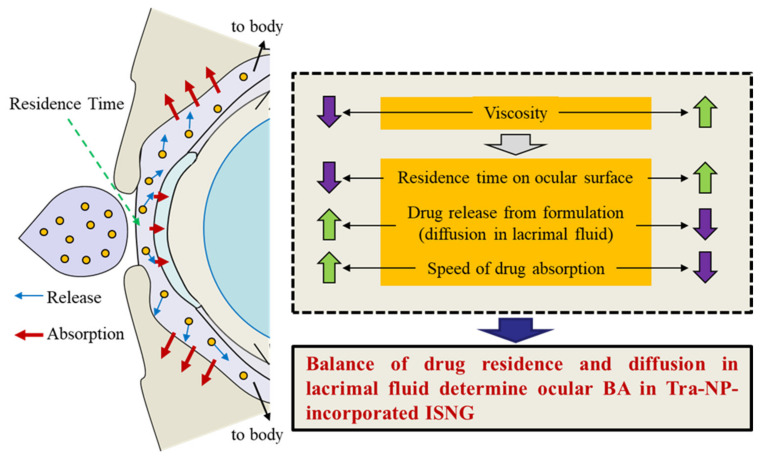
The balance of drug residence time and drug diffusion in lacrimal fluid related to the ocular *BA* of Tra-NP-incorporated ISNGs.

**Table 1 pharmaceutics-13-01425-t001:** Composition of ophthalmic Tra formulations.

Formulation		Treatment (*w*/*v* %)								
		Tra	BAC	Man	HPβCD	F127	MC	F68	Car	
Non-ISG	mTRA	0.5	0.001	0.1	5	-	-	-	-	-
	nTRA	0.5	0.001	0.1	5	-	-	-	-	Bead mill
Low ISG base	nTRA-F127-L	0.5	0.001	0.1	5	10		-	-	Bead mill
	nTRA-MC/F127-L	0.5	0.001	0.1	5	10	3	-	-	Bead mill
	nTRA-F68/F127-L	0.5	0.001	0.1	5	10	-	3	-	Bead mill
	nTRA-Car/F127-L	0.5	0.001	0.1	5	10	-	-	0.2	Bead mill
High ISG base	nTRA-F127-H	0.5	0.001	0.1	5	15	-	-	-	Bead mill
	nTRA-MC/F127-H	0.5	0.001	0.1	5	15	3	-	-	Bead mill
	nTRA-F68/F127-H	0.5	0.001	0.1	5	15	-	3	-	Bead mill
	nTRA-Car/F127-H	0.5	0.001	0.1	5	15	-	-	0.2	Bead mill

The data are expressed as the *w*/*v* %.

**Table 2 pharmaceutics-13-01425-t002:** Physical properties of ophthalmic TL formulations.

Formulation	Particle Size (µm)	NP Number (×10^11^ Particles/mL)	Zeta Potential (mV)	Solubility (mM)	Viscosity (mPa·s)		
					4 °C	20 °C	37 °C, pH6.8
nTRA	103 ± 5.3	10 ± 1.1	–55 ± 0.9	0.30 ± 0.05	1.4 ± 0.1	1.2 ± 0.1	1.1 ± 0.1
nTRA-F127-L	117 ± 4.8	10 ± 0.8	–50 ± 1.3 *	0.55 ± 0.05 *	6.6 ± 0.5 *	5.9 ± 0.6 *	5.7 ± 0.4 *
nTRA-F127-H	110 ± 6.9	10 ± 1.1	–46 ± 1.4 *^,#^	0.77 ± 0.06 *^,#^	20 ± 1.8 *^,#^	16 ± 0.8 *^,#^	65 ± 6.1 *^,#,$^
nTRA-MC/F127-L	108 ± 4.5	9.3 ± 0.6	–56 ± 1.0	0.56 ± 0.05 *	8.9 ± 1.1 *	8.0 ± 1.0 *	12 ± 1.0 *^,$^
nTRA-MC/F127-H	92 ± 5.7	9.0 ± 0.4	–45 ± 0.5 *^,#^	0.78 ± 0.06 *^,#^	87 ± 5.4 *^,#^	76 ± 5.6 *^,#^	119 ± 8.9 *^,#,$^
nTRA-F68/F127-L	102 ± 4.2	8.8 ± 0.6	–52 ± 1.2 *	0.85 ± 0.06 *	8.5 ± 0.6 *	7.6 ± 0.8 *	9.3 ± 0.9 *
nTRA-F68/F127-H	103 ± 7.2	8.2 ± 1.7	–38 ± 1.8 *^,#^	0.87 ± 0.08 *	73 ± 5.3 *^,#^	70 ± 4.9 *^,#^	106 ± 8.7 *^,#,$^
nTRA-Car/F127-L	106 ± 6.4	8.4 ± 1.3	–78 ± 1.1 *	1.62 ± 0.18 *	8.6 ± 0.6 *	7.7 ± 0.6 *	28.8 ± 2.1 *^,$^
nTRA-Car/F127-H	94 ± 7.8	11 ± 1.9	–89 ± 1.8 *^,#^	2.34 ± 0.23 *^,#^	78 ± 5.1 *^,#^	70 ± 5.7 *^,#^	123 ± 7.8 *^,#,$^

The compositions of the ophthalmic Tra formulations are shown in Table 1. Solubility was measured at 20 °C. *n* = 8. * *p* < 0.05 vs. nTRA for each category. ^#^ *p* < 0.05 vs. Tra-NP-incorporated ISNGs with a high ISG base for each category. ^$^ *p* < 0.05 vs. 20 °C for each category.

**Table 3 pharmaceutics-13-01425-t003:** Particle size frequencies of Tra in the reservoir chamber of a methacrylate cell containing each ophthalmic Tra formulation 5 h after treatment.

Formulation	nTRA	nTRA-F127		nTRA-MC/F12		nTRA-F68/F127		nTRA-Car/F127	
		L	H	L	H	L	H	L	H
Particle size (nm)	238 ± 3.2	203 ± 7.7	173 ± 3.5	204 ± 6.1	157 ± 3.4	203 ± 7.0	186 ± 2.3	173 ± 5.8	173 ± 3.7
NP number (×10^9^ particles/mL)	17 ± 0.8	13 ± 1.1	11 ± 0.7	3.5 ± 0.2	2.9 ± 0.2	8.7 ± 0.6	6.3 ± 0.2	9.2 ± 0.9	7.6 ± 0.6

The compositions of the ophthalmic Tra formulations are shown in Table 1.

## Data Availability

Not applicable.

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
