# Peer review of "Balance of Drug Residence and Diffusion in Lacrimal Fluid Determine Ocular Bioavailability in In Situ Gels Incorporating Tranilast Nanoparticles"

_pharmaceutics, 2021, doi:10.3390/pharmaceutics13091425_

Round 1

Reviewer 1 Report

In this manuscript the authors investigate the ocular bioavailability of Tranilast loaded in nanoparticles incorporated in in situ gelling system. The manuscript is well written, and the experimental work is of interest to the readership of this journal. However, I would like to understand the justification of incorporation of the Tra-NPs into ISG? It was mentioned this was done to increase the drug ocular residence time. If that is the case, how to prevent the drug leaching out to the ISG? Why using nanoparticles in the first place why not loading the drug directly into the ISG

I recommend a minor revision providing that the authors give clear justifications/answers to the comment above and the questions below

  • I am not sure about the title of this manuscript. Although the ISG contains nanoparticles in it, surely it is not an In Situ Nano Gels. Please revise the title or clarify
  • Has the HPLC method been validated and if so, according to which guidelines?
  • Can the authors add Images of the HPLC chromatogram to the supplementary materials
  • Why the flow rate is too low, what is the effect on the peak shape?
  • What was the retention time of the analyte?
  • Why not relay on the PDI rather than performing the dispersibility test ?
  • For the Corneal Toxicity assay, the cells were exposed to the treatment for 2 minutes only. The aim of the study to increase the drug residence time. I would imagine you should have exposed the cells for the treatment for longer time.
  • What is the control in the Corneal Toxicity? I can’t see the non-treated on figure 4
  • What is the effect of the drug only on the cell line? Is the toxicity from the formulation or the API?
  • Not sure why is the advantages of having figure 2?

Author Response

We carefully revised our manuscript according to the suggestions of the reviewer 1, and details are as follows.

< Q and A for Reviewer 1>

Q1. It was mentioned this was done to increase the drug ocular residence time. If that is the case, how to prevent the drug leaching out to the ISG? Why using nanoparticles in the first place why not loading the drug directly into the ISG.

A1. Thank you very much for pointing this out. Eyedrops based on ISG systems convert to a gel upon ocular administration, and the enhanced viscosity prevent the lost of eyedrops through nasolacrimal drainage and eye blinking, resulting in enhancement of the drug ocular residence time and drug bioavailability. Therefore, we attempted to design the in situ gels incorporating Tra-NPs. In the preparation of Tra-nanoparticles using bead mill method, the additives are important, and the particle size of Tra was not nano-size when the Tra was milled with ISG base. In order to respond to the reviewer’s comment, we added the content in the Results (line 267-261).

Q2. I am not sure about the title of this manuscript. Although the ISG contains nanoparticles in it, surely it is not an In Situ Nano Gels. Please revise the title or clarify.

A2. The reviewer’s comment is correct. We collected the title to “Balance of Drug Residence and Diffusion in Lacrimal Fluid Determine Ocular Bioavailability in In Situ Gels Incorporating Tranilast Nanoparticles”.

 Q3. Has the HPLC method been validated and if so, according to which guidelines? Can the authors add Images of the HPLC chromatogram to the supplementary materials.

A3. The reviewer’s comments are very important. In order to respond to the reviewer’s comment, we cited the reference for HPLC method of Tra (Ref. 18-20), and added the HPLC chromatogram in the supplementary materials (Figure S1, Reference 18-20).

 Q4. Why the flow rate is too low, what is the effect on the peak shape? What was the retention time of the analyte?

A4. The reviewer’s comment is correct. The peak shape is clear, and the retention time is approximately 6.25 min. We added the HPLC chromatogram in the supplemental data (line 158-159, Figure S1).

Q5. Why not relay on the PDI rather than performing the dispersibility test ?

A5. Thank you for pointing out this. In this study, we attempted to evaluate whether the aggregation and precipitation of Tra-naoparticles was observed for 1 month by using the dispersibility test. In order to respond to the reviewer’s comment, we added the explanation of the dispersion stability test in the Materials and Methods (line 162-164).

Q6. For the Corneal Toxicity assay, the cells were exposed to the treatment for 2 minutes only. The aim of the study to increase the drug residence time. I would imagine you should have exposed the cells for the treatment for longer time.

A6. The reviewer’s comments are very important. In general, the eye drops was diluted by the lacrimal fluid (approximately 10-fold dilution) immediately after the instillation, and drugs is flowed through the nasolacrimal duct. On the other hand, we used the non-diluted eye drops in this in vitro study, and treated for 2 min. Therefore, it was suggested that the stimulation in the in vitro study was higher than that in the in vivo condition. Taken together, the stimulation time was determined at 2 min. On the other hand, it is important to investigate the effect of prolonged drug residence time on the corneal toxicity. Therefore, we also evaluated the corneal toxicity in the in vivo study (Figure 4B). In order to respond to the reviewer’s comment, we added the content in the Materials and Methods (line 179-181, 190-192).

Q7. What is the control in the Corneal Toxicity? I can’t see the non-treated on figure 4. What is the effect of the drug only on the cell line? Is the toxicity from the formulation or the API?

A7. Thank you for pointing out this. The Abs of non-treatment group was used as Control, and the cell viability (%) was calculated as a ratio [Abstreatment/Absnon-treatment (Control)×100]. In this study, the cell toxicity reflect the corneal damage from formulation. In order to respond to the reviewer’s comment, we added the content in the Materials and Methods (line 188-189).

Q8. Not sure why is the advantages of having figure 2?

A8. Thank you very much for pointing this out. Figure 2 showed the particle size frequencies of Tra-NP-incorporated ISNG. If this data is removed, it is not proved whether the Tra particles maintain nano-size in the dispersions containing ISG additives. Thank you for pointing out this.

Thank you for great comments.

Reviewer 2 Report

Dear Authors,

your manuscript "Balance of Drug Residence and Diffusion in Lacrimal Fluid Determine Ocular Bioavailability in In Situ Nano Gels Incorporating Tranilast Nanoparticles" is very interesting. The experimental plan reflect the purpose of the objectives you have established.   Therefore, your manuscript will be  accepted in present form.

Best regards

Author Response

We carefully revised our manuscript according to the suggestions of the reviewer 2, and details are as follows.

< Q and A for Reviewer 2>

Q1. Your manuscript "Balance of Drug Residence and Diffusion in Lacrimal Fluid Determine Ocular Bioavailability in In Situ Nano Gels Incorporating Tranilast Nanoparticles" is very interesting. The experimental plan reflect the purpose of the objectives you have established. Therefore, your manuscript will be accepted in present form.

A1. Thank you very much for great comments. We are very grateful for your review.

Reviewer 3 Report

Manuscript IDpharmaceutics-1365586

Title: Balance of Drug Residence and Diffusion in Lacrimal Fluid Determine Ocular Bioavailability in In Situ Nano Gels Incorporating Tranilast Nanoparticles

Recommendation: Major revision

In this study, the authors synthesized in situ nano gels incorporating tranilast nanoparticles (Tra-NP-incorporated ISNGs) for the high uptake and release into ophthalmic tissues and its preventive effect on inflammatory mediators in conjunctiva inflammation. This work is interesting, but the theory and the experiments are not sufficient. There are still some issues need to be addressed as follows,

  1. In introduction part, there is a lack of other studies on the use of TRA for eye diseases. The reviewers want to obtain more explanations on this issue. Please dig the literature and highlight the advantages of this paper over previous studies.

  1. According to the results of Figure 1B, the size of Tra-nanoparticles is (79±14) nm. However, in the Figure 1C, the size of Tra-nanoparticles is change to (103.7±5.3) nm, please explain this phenomenon. Besides, the described for the meaning of AFM images (Figure 1D and Figure 2) is not found in the paper. The reference of LPS-induced conjunctiva inflammation rat model is highly recommended to be provided.

  1. In Figure 3A, we can observe that TRA concentration at 4 °C is obvious lower than 20 °C according to the images of nTRA dispersibility, which is contrary to Figure 3B, please check and correct it.

  1. The detailed morphology of Tra-NPs cannot be observed in this work, leading to the integrity of prepared Tra-NPs is doubtful.

  1. In Table 3, after incubated in the reservoir chamber of a methacrylate cell for 5 h, the particle size of TRA has been became microparticles, indicating the NPs cannot stable enough to treat ophthalmic diseases effectively.

  1. In Figure 4B, the differences of formulations cannot be observed clearly from the qualitative images, especially for CA-TRA, MC/F127-L, Car/F127-H and Car/F127-L, the quantitative data and relative demonstrating are vital to understand the therapy effect for formulations on the rat cornea.

  1. To truly understand of the properties of Tra-NPs, the author should elucidate qualitative analysis at cellular level of NPs in preventing the conjunctiva inflammation effectively and revealing the specific mechanism of anti-inflammation in cornea.

  1. The manuscript would benefit from careful read through. There are a number of grammatical errors in the manuscript such as line 55 “because of their excellent potential”, line 247 “Fig. 1” should be changed as “Figure 1”, line 380 “retention” should be changed as “retention time”. The authors should strictly revise the language of this manuscript.

Author Response

We carefully revised our manuscript according to the suggestions of the reviewer 3, and details are as follows.

< Q and A for Reviewer 3>

Q1. In introduction part, there is a lack of other studies on the use of TRA for eye diseases. The reviewers want to obtain more explanations on this issue. Please dig the literature and highlight the advantages of this paper over previous studies.

A1. Thank you very much for pointing this out. In order to respond to the reviewer’s comment, we added the information for other studies on the use of Tra for eye diseases (line 85-88, reference 36-40).

Q2. According to the results of Figure 1B, the size of Tra-nanoparticles is (79±14) nm. However, in the Figure 1C, the size of Tra-nanoparticles is change to (103.7±5.3) nm, please explain this phenomenon. Besides, the described for the meaning of AFM images (Figure 1D and Figure 2) is not found in the paper.

A2. The reviewer’s comments are very important. The particle size in the Figure 1B was measured by a laser diffraction particle size analyzer SALD-7100. On the other hand, a Dynamic Light Scattering QuantumDesign NANOSIGHT LM10 was used to measure the particle size in the Figure 1C. The difference in particle size is due to these measuring method. In addition, we also showed the AFM image of Tra. The particle size of Tra was similar the data of SALD-7100 and NANOSIGHT LM10, and the Tra-NPs was a shape close to spherical. The measurements by using the multi-methods provide more detail morphology of Tra-nanoparticles. In order to respond to the reviewer’s comment, we added the contents in the Results (line 265, 297-298).

Q3. The reference of LPS-induced conjunctiva inflammation rat model is highly recommended to be provided.

A3. In order to respond to the reviewer’s comment, we added the reference of LPS-induced conjunctiva inflammation rat model (Reference 41,42).

Q4. In Figure 3A, we can observe that TRA concentration at 4 °C is obvious lower than 20 °C according to the images of nTRA dispersibility, which is contrary to Figure 3B, please check and correct it.

A4. The reviewer’s comments are very important. The dispersibility of ophthalmic formulations containing Tra-NPs (Figure 3B-F) were evaluated by the measurement of the Tra levels in the sample collected from the upper 90% of the tube. The Tra contents in the upper 90% of the tube at 4 °C tend to be higher than that at 20 °C. Thank you for pointing out this.

 Q5. The detailed morphology of Tra-NPs cannot be observed in this work, leading to the integrity of prepared Tra-NPs is doubtful.

A5. Thank you for pointing out this. We measured the particle size and distribution using the 2 method (a laser diffraction particle size analyzer SALD-7100 and a Dynamic Light Scattering Quan-tumDesign NANOSIGHT LM10). In addition, the form of Tra was also evaluated using AFM image, and the particle size of Tra was similar the data of SALD-7100 and NANOSIGHT LM10, and the form of Tra-NPs was spherical. Moreover, we measured the diffusion behavior of Tra in the ophthalmic Tra-NP-incorporated ISNGs, and demonstrated the corneal toxicity and the behavior in the ophthalmic tissue of rats instilled with the Tra formulations. We considered that the analysis by these multiple methods express the detailed morphology of the drug. Thank you very much for pointing this out.

 Q6. In Table 3, after incubated in the reservoir chamber of a methacrylate cell for 5 h, the particle size of TRA has been became microparticles, indicating the NPs cannot stable enough to treat ophthalmic diseases effectively.

A6. The unit is wrong. We collected the unit to “nm” from “µm”. Thank you for pointing out this. (Table 3).

 Q7. In Figure 4B, the differences of formulations cannot be observed clearly from the qualitative images, especially for CA-TRA, MC/F127-L, Car/F127-H and Car/F127-L, the quantitative data and relative demonstrating are vital to understand the therapy effect for formulations on the rat cornea.

A7. The reviewer’s comments are very important. We analyzed the corneal wound area by using Image J, and the wound area of rat repetitive instilled with nTRA-Car/F127-H was 0.71 ± 0.13 mm2 (n=8). In order to respond to the reviewer’s comment, we added the data in the Results (line 196, 351-352).

Q8. To truly understand of the properties of Tra-NPs, the author should elucidate qualitative analysis at cellular level of NPs in preventing the conjunctiva inflammation effectively and revealing the specific mechanism of anti-inflammation in cornea.

A8. The mechanism of anti-inflammation in Tra have already reported previous study, and we showed that the changes in EB exudation, NO levels and TNF-α levels in the conjunctiva of conjunctivitis rats 5 h after the instillation of each ophthalmic Tra formulation (Figure 8). These data showed that the enough Tra was taken into cells from the formulations containing Tra-nanoparticles. On the other hand, it is important to elucidate qualitative analysis at cellular level of nanoparticles. Therefore, we added the importance in the Discussion. Thank you for pointing out this (line 561-563).

 Q9. The manuscript would benefit from careful read through. There are a number of grammatical errors in the manuscript such as line 55 “because of their excellent potential”, line 247 “Fig. 1” should be changed as “Figure 1”, line 380 “retention” should be changed as “retention time”. The authors should strictly revise the language of this manuscript.

A9. Thank you very much for pointing this out. In order to respond to the reviewer’s comment, we collected the grammatical errors in the manuscript.

 Thank you for great comments.

Round 2

Reviewer 3 Report

The manuscript has been revised mostly according to the comments and suggestions of the reviewers,  and the manuscript can be accepted as it is.